# OpenReview forum: "AdvPrefix: An Objective for Nuanced LLM Jailbreaks"
_NeurIPS.cc/2025/Conference — NeurIPS 2025 poster_

### Official Review · Reviewer_r6wM · 2025-07-01

**Clarity:** 3
**Significance:** 4
**Originality:** 3
**Rating:** 5
**Confidence:** 4

**Summary:**

This paper proposes a plug-and-play, adaptive prefix-forcing objective for traditional jailbreak attacks (e.g., GCG), aiming to address the two defects of misspecified and overconstrained in traditional attacks. The authors use a meta-evaluation framework to analyze the problem of overestimation of ASR in existing evaluation methods, especially in newer and more powerful models where incomplete or unfaithful responses are mistakenly recorded as successful attacks. Secondly, by defining two criteria for prefix target selection, high prefilling ASR and low initial NLL, an automated prefix selection method is proposed. Related experiments verify that existing attacks can greatly improve ASR on multiple LLMs when AdvPrefix is applied.

**Questions:**

- During the optimization process of the Multi-prefix objective, are there conflicts between different prefixes?
- In the candidate generation step, how is rule-based construction implemented?
- For the Gemma-2 model, some effective attack prefixes do not begin with an affirmative response (for example, prefixing directly with “Password Cracking Algorithm”). Can authors elaborate on the success of such “non-affirmative” prefixes?

**Ethical Concerns:**

["NO or VERY MINOR ethics concerns only"]

**Final Justification:**

The author's response addressed my concerns, and I decided to maintain my positive score.

**Limitations:**

yes

**Quality:**

4

**Strengths And Weaknesses:**

## Strengths
- **Generalizable contribution.** This paper aims at a fundamental component shared by many attack algorithms - the optimization goal. This perspective makes its contribution very general. As a plug-and-play module, AdvPrefix can improve the performance of a large number of existing attack frameworks and has high value.
- **Thorough analysis of the problem.** Through rigorous meta-evaluation and innovative "prefilling attack" experiments, the authors convincingly quantified and demonstrated the two major flaws of traditional optimization objectives: "misspecified" and "overconstrained". This solid problem provides a new research perspective for subsequent attack methods.
- **The method is cleverly designed.** AdvPrefix is very cleverly designed, and the two selection criteria it proposes (high pre-filled ASR and low initial NLL) form a perfect correspondence with the two problems it aims to solve (misspecified and overconstrained). The entire prefix generation and screening process is designed as an automated pipeline with clear ideas and rigorous logic.
## Weaknesses
- **Dependence on Uncensored Models.** AdvPrefix relies on uncensored LLMs to generate prefixes. Does the gain of the attack depend on the choice of the uncensored model? If a larger model (e.g., 70B) is chosen, is it beneficial to the attack effect? The author did not discuss this.
- **The selection process is unclear.** The author claims that the candidate prefixes are selected by taking the weighted sum of two criteria to select the top $k$ prefixes. However, it is not clear how to obtain the weighted sum and the setting of $k$. In addition, what impact do they have on the effectiveness of the attack?
- **Some minor typos.** Such as “model's” in Line 36 should be “models'”, and there is an extra "." in Line 135.

---

> ### Author Rebuttal · Authors · 2025-07-31
>
> Thank you for the encouraging feedback!
>
> **Weakness 1.**
> We can build the candidate prefix pool either with rule-based construction (by observing model output patterns) or with guided decoding using uncensored models. These uncensored models can be pretrained base models, helpful-only models, malicious-data–finetuned models, or models with mechanistic refusal suppression. The attack’s gain empirically depends on how good the candidate prefixes are under our two criteria, rather than on which uncensored model generates them. Using a larger model does not necessarily improve the attack.
>
> **Weakness 2.**
> We choose the sum weight and k based on a small-scale run and keep them fixed across all experiments. We did not fine-tune these values because the current results already demonstrate the effectiveness of our objective. Ideally, the sum weight could be made optimization-method–dependent, since stronger methods like GCG can prioritize high prefilling ASR and tolerate higher NLL.
>
> **Question 1.**
> They do not conflict. By using the log of the sum of probabilities instead of the sum of log probabilities, our multi-objective is intentionally designed to allow "arbitrarily choose 1 out of k" rather than "choose all k with a specific mixture (distribution)."
>
> **Question 2.**
> The rule-based construction simply uses "Sure, here is …" style prefixes (from GCG) to diversify our candidate pool. A possible future improvement is to generate such rules automatically by recognizing patterns in the target model’s responses.
>
> **Question 3.**
> These non-affirmative responses are automatically generated and selected by our pipeline. Once triggered by the attack prompt, the target model often continues naturally, e.g., "Password Cracking Algorithm: \*\*Step 1 …." We find that Gemma-2 in particular tends to favor non-affirmative responses compared to other models, which our method exploits.

---

> > ### Comment · Reviewer_r6wM · 2025-08-03
> > **Response to Authors**
> >
> > Thank you for your response. I decide to keep my positive score.

---

### Official Review · Reviewer_tzJB · 2025-07-03

**Clarity:** 1
**Significance:** 2
**Originality:** 2
**Rating:** 3
**Confidence:** 5

**Summary:**

This paper proposes AdvPrefix, a prefix-forcing objective intended to raise jailbreak attack-success rate (ASR).
For each malicious request the authors

1. sample a pool of candidate prefixes with an “uncensored” LLM,
2. (implicitly) filter obvious refusals,
3. score each prefix by ASR and –NLL and keep the top-K prefixes for use with gradient-based attacks such as GCG and AutoDAN.
4. On Llama-2-7B-chat, Llama-3-8B, 3.1-8B-Instruct and Gemma-2-9B-it this increases ASR: For example from 16 % to 70 % on Llama-3-8B.
5. The authors also release 800 human-labelled outputs and a refined Llama-3.1-70B judge.

**Questions:**

If the authors will address at least some of the key issues (see points 1-7 in Weaknesses), I would see it as a major improvement making me reconsider the score. Addressing other concerns (see points 8-9 in Weaknesses), would further help me increase the score upon the resolution of points 1-7.

**Ethical Concerns:**

["NO or VERY MINOR ethics concerns only"]

**Final Justification:**

Because of the lacking crucial details, comparisons, and experiments in the paper I maintain my negative ranking. The rebuttal of the authors unfortunately did not address my core concerns.

**Limitations:**

yes, however the limitation of strong assumption of prefiling is not mentioned.

**Paper Formatting Concerns:**

I did not observe major formatting issues.

**Quality:**

2

**Strengths And Weaknesses:**

## Strengths:

1. Addresses a timely safety problem: vanilla jailbreak objectives often yield partial or unfaithful harmful outputs.
2. Proposes a simple improvement for jailbreaking attacks by optimizing the objective over the set of candidate prefixes.
3. Gains across two attacks and four open-source LLMs.

## Weaknesses:

### Key weaknesses:

1. Semi-manual, vaguely described prefix search: The “prefix-selection pipeline” relies on heuristically prefilling uncensored LLM responses with words like “Here”, “To”, “Sure”. Details such as candidate pool size, rejection rules and weighting of ASR vs NLL are missing, making the method hard to reproduce.
2. Limited generalization evidence to other attacks and models: Results stop at 9 B-parameter open models and two gradient-based attacks; prior work shows a single handcrafted template can transfer to a wide range of models ([3]) and that fluency constraints can be added automatically ([4]) to any attack; AdvPrefix shows no such cross-model or cross-attack generalization.
3. Strong pre-fill assumption: The approach assumes the attacker may prepend text inside the user/system prompt. Hosted APIs often disallow this ([3]).
4. Selecting "easy" prefixes resembles reward-misspecification or multi-objective jailbreaks [1, 2]. Thus, there is no significant algorithmic contribution.
5. Unexplained drop with longer prefixes: Figure 5 shows that on Llama-3.1 the win-rate falls when the prefix budget grows from 20 to 40 tokens, showing that there might be issues with optimization as bigger search space should allow for better solutions. The paper gives no analysis or ablation on this matter.
6. Undefined success metric: The paper replaces standard ASR with a "Successful Attack" measure but never defines it, nor justifies why plain ASR is insufficient.
Judge dependency, no human agreement.
7. This approach is time-costly due to the need of mining the set of prefixes and is not realistic for large-scale red-teaming.

### Other weaknesses:
8. The paper does not use the full 9 pages such that ending looks abrupt.
9. In Appendix, L 909, the authors write "To handle cases where uncensored LLMs still refuse highly harmful prompts, we prefill their responses with phrases like “Here”, “To”, or “Sure” to ensure compliance. Note that we **can** also construct candidate prefixes using rule-based methods or with base (non-instruction-tuned) LLMs instead of uncensored LLMs." This contradicts the statements in the paper that prefill is mostly done with uncensored LLMs, which causes confusion.

--------

## References

[1] "Jailbreaking as a Reward Misspecification Problem". ICLR 2024.

[2] "Fluent Student-Teacher Red-Teaming". arXiv 2024.

[3] "Jailbreaking Leading Safety-Aligned LLMs with Simple Adaptive Attacks". ICLR 2025.

[4] "An Interpretable N-gram Perplexity Threat Model for Large Language Model Jailbreaks". ICML 2025.

---

> ### Author Rebuttal · Authors · 2025-07-31
>
> Thank you for clearly listing your concerns! We believe there are several misunderstandings.
>
> 1. **Prefilling uncensored LLM responses is completely *optional***. We do this because (1) the specific open-source uncensored model we use still refuses to answer some highly harmful questions, but this is not an issue for some other uncensored (closed-source) models; (2) prefilling diversifies the candidate pool, which never hurts the final prefix selection. In fact, many of our released prefixes (especially on Gemma-2 9B) start with arbitrary words like "**\*\*Phase 1: Reconnaissance\*\***" or "**\*\*Subject:\*\* Urgent: Account Security Update Required**".
>
>     Details for candidate pool size, rejection rules, and weighting, are included in our codebase, which we will discuss them in more detail in the Appendix. The result is reproducible using our released code.
>
> 2. Results stopping at 9B model and not addressing cross-model transferability are indeed scope limitations of our work. Our method itself is not restricted by this and applies whereever "Sure, here is" appears.
>
> 3. We address white-box settings, where attackers have access to logprobs and can do inference based on prefilled text. This does not extend to black-box settings, but is useful for model developers to red-team and adversarially train their models.
>
> 4. **Our method is *fundamentally different* from [1] and [2]**. Paper [1] still uses "Sure, here is ..." as positive (harmful) answers, whereas [2] does not use prefix-forcing objective at all. Our related work section has discussed these two papers.
>
> 5. **The pipeline of "our objective $\rightarrow$ random GCG optimization $\rightarrow$ defender response sampling $\rightarrow$ baseline model sampling $\rightarrow$ preference judge sampling" contains intrinsic randomness and our results already show statistically-significant improvements.** Optimization contains randomness, especially for search-based method like GCG; jailbreaking is also not perfectly gradable, especially for win-rate computation in figure 5 using our preference LLM judge and a baseline model. Our objective already shows 40%+ absolute win-rate improvement over the original objective in that group, but we cannot guarantee to always have absolute improvement when extending 20 attack prompt tokens to 40.
>
>
> 6. **We use both standard ASR and pair-wise response quality comparison** (win-rate over baselines), and **the "successful attack" is simply just standard ASR**. Our entire section 2, figure 2, figure 6, and table 4, show why we need better judge. "No human agreement" $\rightarrow$ our Table 4 reports human agreement rates of different judges, where all the labeling data are released.
>
> 7. We discuss the computational cost in section 5.2. Getting a final selected prefix requires 5 minutes running on an A100 GPU, where the VRAM is mostly used for a 70B reasoning judge (can be replaced by GPU-free API call). **Our method is scalable and infra-friendly since selecting the prefix only requires inference or computing logprobs**, without gradient computation or optimization.
>
> 8. Our paper is just **three lines** short of a **full nine pages**.
>
> 9. This point is similar to point 1. Note that we can generate seed prefixes with just uncensored LLMs, and our heuristics are just for diversifying the candidate pool which helps imperfect uncensored LLMs and never hurts prefix selection.

---

> ### Comment · Reviewer_tzJB · 2025-08-06
>
> Dear authors,
>
> thank you for taking your time to answer all of my concerns. Please find my responses to each of them below and correct me, if I missed something.
>
> 1.
> - **Prefilling.** I have carefully looked at your response and the paper, however, if I understand it correctly, prefilling is necessary for your algorithm: In your abstract, Section 1, Figure 4, Sections 4.2, Section 4.3, Section 5, and Section 7 you write that it is one of 2 necessary criteria. Should this not be the case, it would be nice to have a clarification regarding this.
>
> -  **Details about the algorithm.** I would appreciate having a clear description of the algorithm. This is because i) it is the main contribution of your paper, ii) currently there is some confusion about how it works and I find it critical: a) it is not clear, how the prefilling is used, especially because you need to evaluate prefill ASR (see the point 1 above); b) it is not clear, what the pool of candidates is, how big it is, how it is constructed (per model per request); c) what is the weighting used for your prefilling-ASR and NLL objectives; d) which manual prompt you are using (mentioned in L145-147 and mentioned to correlate with actual jailbreak ASR in Figure 7 despite r-value being low) and how; e) rejection rules; f) how exactly your judge is configured and used including its system prompt; g) you mention that you consider two very different threat models in Section 5, but do not specify in experiments which you are using.
>
> - **Lacking code.** Thank you for pointing me to your code, even though I would have preferred to see the important details in the paper. I have discovered that it i) generates prefixes for only 2 behaviors and only 1 model and does not run the attack itself, thus I cannot judge if the algorithm can work and how general it is or if it has been overfitted on the two prompts; ii) uses the wrong configuration of HarmBench judge (please refer to https://github.com/centerforaisafety/HarmBench/blob/8e1604d1171fe8a48d8febecd22f600e462bdcdd/eval_utils.py#L309 for the correct configuration comparing it to yours in HarmBenchEvaluator class of scorer_pasr.py).
>
> 2. **Limitations.** I appreciate that you admit your limitations. However, i) I would appreciate if you would discuss them in your paper as well as these are significant limitations; ii) this highlights in my opinion the non-generalizability of your proposed solution when comparing to [1], where the authors have one proposed template that works across both white- and black-box models of different size including different versions of Claude and GPT models with $100%$ ASR. Could it be the case that having such a template does not require a complicated prefix search? Did you compare your attack with [1] given that you are citing it?
>
> 3. **Limitation - white-box setting.** Allowing for only white-box attacks indeed might be useful and novel for testing the models by model developers. However, I would presume that both model developers and attackers would prefer Pareto-optimal attacks in FLOPs and perplexity (see for example [2]). I would be curious, if your attack (or its computation overhead to an attack) is Pareto-optimal compared to wide range of existing strong attacks such as [1]?
>
> 4. **Related work clarification.** Thank you for your clarification on this matter.
>
> 5. **Unexplained drop in ASR when increasing the length of the suffix.** I think this is a critical concern because as you show in Figure 9 and because there is no algorithm provided I assume that the model sees only the adversarial suffix and there is no behavior visible to the model in the request. This is a very unusual threat model and thus requires special care an analysis in my opinion.
>
> 6. **Judge.** Thank you for your clarification. Upon having a closer look, I see that you indeed user agreement in Table 4 in Appendix. However, i) having the description of the user study conducted is important for understanding it and I cannot find it; ii) your implementation is wrong for HarmBench judge (see the point 1 above); iii) it is suspicious that in Table 4 JailbreakBench judge performs worse than HarmBench, when in [3], Table 1, it is the opposite; iv) Llama Guard 1,2,3 are excluded in Table 4.
>
> 7. **Cost.** 5 minutes per prefix per prompt per model is a lot in my opinion and adds non-trivial computation cost. Is it Pareto-optimal (see point 3 above)?
>
> 8. **Missing 3 lines.** I agree with it, however such things show the attention to details and are important to address in paper in my opinion. Please note that it is only a minor weakness.
>
> 9. **Prefix construction.** Please refer to point 1 above.
> --------
>
> **References:**
>
> [1] "Jailbreaking Leading Safety-Aligned LLMs with Simple Adaptive Attacks". ICLR 2025.
>
> [2] "An Interpretable N-gram Perplexity Threat Model for Large Language Model Jailbreaks". ICML 2025.
>
> [3] "JailbreakBench: An Open Robustness Benchmark for Jailbreaking Large Language Models". NeurIPS 2024.

---

### Official Review · Reviewer_zhoh · 2025-07-04

**Clarity:** 3
**Significance:** 3
**Originality:** 4
**Rating:** 3
**Confidence:** 4

**Summary:**

The paper introduces AdvPrefix, mainly a new objective designed to improve the effectiveness of jailbreak attacks on LLMs. The objective selects model-dependent prefixes based on two criteria: high success rates when used to prefill responses and low negative log-likelihood (NLL), making them both effective and easy to elicit. The method can be fairly easily integrated into frameworks such as GCG. The empirical results only show on the smallest tier of models though.

**Questions:**

1. While the paper is very interesting, it probably needs a better scope of experiments, for examples, only evaluating models at 7B-tier might be concerning. Although I understand that the authors might be limited by the computing in evaluating a bigger model, but unfortunately, it's probably necessary to have a more justified evaluation of the proposed method.

2. While the objective design is well principled, the experiment section seems to drop this pricinple. For example, in the experiments, what is the threshold of NLL are considered as low NLL to be selected. Will this threshold matter in the final performances and how?

3. Similarly, as the prefix is the essential part of the method, generating the prefix needs a much more comprehensive ablation study.

4. Which benchmark are the experiments conducted on? This principled method might deserve a much more stronger benchmark to demonstrate its performances [1], where many existing attack algorithms fail.

[1] Jailbreaking large language models against moderation guardrails via cipher characters

**Ethical Concerns:**

["NO or VERY MINOR ethics concerns only"]

**Final Justification:**

I appreciate the authors offer a friendly and honest rebuttal, and the authors agree with many concerns I raised. It would be more helpful if the authors can offer some updated results though.

**Limitations:**

Yes.

**Paper Formatting Concerns:**

None.

**Quality:**

3

**Strengths And Weaknesses:**

strengths:

- The objective design is very well principled and justified, prefixes are selected using measurable, interpretable criteria.

- Overall, a very simple method, yet have strong performances.

weakness:

- It's a bit concerning that the papers later only use the authors evaluation criteria to evaluate, while the authors have reasons to believe their evaluation is superior, it might be necessary to evaluate with standard evaluation procedure to let the community inspect.

- the paper probably need a more broader scope of experiments.

---

> ### Author Rebuttal · Authors · 2025-07-31
>
> We appreciate your constructive feedback, especially your recognition that our objective design is principled and your concern about experiment scale. We hope our response helps clear up some of the potential misunderstandings.
>
> **Weakness 1.**
> (1) We do not claim that our evaluation is “superior” in any sense, since harmfulness judgment is inherently subjective. Our labeling for most safety violation categories follows a public benchmark [1], and we release our 800 labeled samples for public inspection.
> (2) We also evaluate our results using standard procedures (HarmBench, JailbreakBench, StrongReject) in Table 3 (Appendix), and report human agreement analysis on these in Table 4 (Appendix).
> (3) Our method is disentangled from any specific labeling criteria and can work with any provided grader (judge).
>
> **Weakness 2 and Question 1.**
> We agree that a broader experimental scope would make our results more comprehensive. We would like to point out, however, that our main computational limitation comes from the computational cost of the selected optimization-based attacks, rather than our prefix generation. Prefix generation only requires inference and log-prob computation, which is scalable and infra-friendly.
>
> **Question 2.**
> This is simply a trick to save computation (by pruning). One can skip this preprocessing if they have better inference throughput. We chose the threshold based on our compute budget. Running without this preprocessing on Llama-3-8B produces similar results because our NLL threshold is well above the typical selected prefixes.
>
> **Question 3.**
> We agree that our work would benefit from a more comprehensive ablation of prefix generation variants. Our current results suggest that we should generate as many diverse candidates as our compute budget allows, and that increasing the size of the candidate pool never hurts.
>
> **Question 4.**
> We use HarmBench, JailbreakBench, StrongReject, our refined judge, and our preference judge on chat-based jailbreak problems without system-level moderation. We agree that our work would benefit from an even stronger benchmark, and testing our method on benchmarks with system-level moderation [2] is a very interesting future direction that we will discuss in the paper!
>
> References
>
> [1] Introducing v0.5 of the AI Safety Benchmark from MLCommons
>
> [2] Jailbreaking large language models against moderation guardrails via cipher characters

---

> > ### Comment · Reviewer_zhoh · 2025-08-05
> > **re rebuttal**
> >
> > I appreciate the authors efforts in responses, while the authors generally agree with the comments, I wonder if the authors will be able to offer the discussions in more details.
> >
> > For example, the author mentions this
> >
> > >"Running without this preprocessing on Llama-3-8B produces similar results because our NLL threshold is well above the typical selected prefixes."
> >
> > it seems the authors have tested and have relevant results, it will be more helpful if the authors post these results directly here.

---

### Official Review · Reviewer_VQ8d · 2025-07-06

**Clarity:** 3
**Significance:** 3
**Originality:** 3
**Rating:** 5
**Confidence:** 4

**Summary:**

This paper provides a more flexible approach to selecting a response prefix for a LLM jailbreak optimization. The approach first generates a list of candidate prefixes (using an uncensored LLM or heuristics), then selects the best prefix using a weighted combination of how good each prefix is based on -- (1) when prefilled in the response how good is the attack success rate, and (2) the perplexity of the prefix given the input prompt. The evaluation shows that AdvPrefix improves ASR for two existing attacks.

**Questions:**

1. It would be useful to add an analysis of the initial perplexities of the candidate response prefixes.
2. Table 1 suggests that adding multiple prefixes to the loss does not help a lot. Is this because the loss objective is not able to simultaneously accommodate multiple prefixes or there is a large disparity between the top and second best prefix?

**Ethical Concerns:**

["NO or VERY MINOR ethics concerns only"]

**Final Justification:**

The discussion helped answer my questions and I remain positive about the paper and increase my rating to 5.

**Limitations:**

yes

**Quality:**

3

**Strengths And Weaknesses:**

Strengths:
1. Tackles an important problem that is often overlooked in existing jailbreak attacks.
2. Extensive evaluation demonstrating support for all design decisions for the proposed attack.

Weakness:
1. Not a major weakness, but the response prefix ranking relies on the assumption that prefilling works to a certain degree (otherwise ASR for all candidates would be equal to zero), and there is a disparity in the initial perplexity of the different candidates. While evidence shows that these assumptions hold for the LLMs evaluated in this paper, they may not be true for all LLMs (which otherwise could still be prone to jailbreaks).
2. Prefilling requires white box access or atleast a more direct access than is provided by most black box LLMs (Gpt4, Gemini et). In that case, the only way for this approach would be to transfer from an open source model. Since the candidate selection is model specific, it is unclear whether the selected response prefix would be any better than a random or pre-selected one.

---

> ### Author Rebuttal · Authors · 2025-07-31
>
> Thank you for the insightful feedback!
>
> **Weakness 1 and Question 1.**
> (1) Prefix ranking, which is essentially predicting extremely rare LLM behaviors, can be NP-complete if we view it as "is there any prompt that makes the model output some certain response p" without any assumptions (e.g., [1]). To make this tractable, we assume a correlation between prefilling ASR and final ASR, which we empirically test in figure 7 (Appendix). We also hypothesize that low initial loss (low initial perplexity) makes optimization easier, leading to lower final loss (lower final perplexity), which we visualize in figure 5 (left). Beyond these individual analyses, our main results demonstrate the end-to-end effectiveness of our method. (2) We agree that validating these assumptions on more LLMs would make our result even stronger!
>
> **Weakness 2.**
> Our work focuses on white-box settings, aiming to help model developers red-team and adversarially train their models. Not addressing black-box transferability is indeed a limitation of our current scope.
>
> **Question 2.**
> Great question! This is because of the large disparity between the top and the second best prefix. We find that using simple heuristics to augment the top prefix (e.g., word replacement) can often lead to better-ranked candidates and better multi-objective results than our current pipeline. We do not use these heuristics to keep the pipeline simple for clarity, but more systematic top-prefix augmentation would be an interesting direction for future work!
>
> Reference
>
> [1] DNN Verification, Reachability, and the Exponential Function Problem

---

> > ### Comment · Reviewer_VQ8d · 2025-08-06
> >
> > Thanks to the authors for their response. I agree with the answers and keep a positive score. I would encourage the authors to add a section to clearly state the assumptions required for the method to work (ex - prefilling assumption), and also the limitations (limited to white-box).

---

### Decision · Program_Chairs · 2025-09-17

**Decision:**

Accept (poster)

**Comment:**

This paper proposes the AdvPrefix, which achieves the prefix-forcing objective that selects one or more model-dependent prefixes to perform jailbreak attacks. The strengths and the weaknesses are clear. Regarding the strengths, the experiments and performance meet the reviewers' expectations. Regarding the weaknesses, the biggest shortcoming lies in its application limitations and security model design. 2 Reviewers suggest rejecting this manuscript due to the white-box adaptability and the insufficient novelty. In my opinion, as an attacking method, white-box settings are always powerless in practical scenarios, and the threat model of this study is not clearly stated. As a red-team testing, this study could not provide guided enhancement suggestions due to its technical design, i.e., the study neither answers the reason why LLMs could be jailbroken nor gives insights into the model decision mechanism.